# The anti-obesogenic metabolite, Lac-Phe, is elevated by metformin treatment in prostate cancer patients

Marijo Bilusic [ID] [1,2 ✉], Durga Prasad Gannamedi[3], Bhipasha Challu [ID] [4], Shomita Ferdous[5], Beatriz Mateo-Victoriano [ID] [5,6], Sheela Pokharel[5], Defne Bayik [ID] [2,7], Janaki Sharma [ID] [1,2], David B Lombard [ID] [2,3,8 ✉] & Priyamvada Rai [ID] [2,5 ✉]

## Abstract

Randomized clinical trials have highlighted the importance of weight control and exercise in improving cancer outcomes, yet the molecular mediators of these benefits remain poorly understood. Recent studies in Type 2 diabetic patients reported that the insulin-sensitizing drug, metformin, exerts its anorexigenic and anti-obesogenic effects through elevation of N-lactoyl-phenylalanine (Lac-Phe), a metabolite also reported to underlie the metabolic benefits of exercise in healthy individuals. Because metformin can be repurposed for oncology, we investigated whether the metformin/Lac-Phe link extends to cancer by profiling serum Lac-Phe levels in non-diabetic prostate cancer patients from BIMET-1, a prospective metformin trial, as well as those receiving metformin for metabolic dysfunction. Irrespective of disease stage, anticancer outcomes, or hormone therapy status, metformin treatment significantly and consistently increased serum Lac-Phe in these cancer patients to concentrations that approximated levels reported after strenuous exercise. Moreover, patients on the metformin arm of BIMET-1 exhibited improved weight management following anti-androgen therapy relative to those in the control group. By generalizing the metformin/Lac-Phe axis, our study provides a new molecular context for the metabolic benefits of repurposing metformin in cancer patients.

**Keywords** Metformin; Lac-Phe; Weight Management; Hormone Therapy; Cancer
**Subject Categories** Cancer; Metabolism

## Introduction

Cancer is the leading cause of mortality worldwide, with over 20 million new cases and over 9 million associated deaths reported in 2022 (Bray et al, 2024). Being overweight or obese is a known cancer risk factor and is associated with worse disease and treatment outcomes (Calle et al, 2003; Lauby-Secretan et al, 2016; Petrelli et al, 2021). Conversely, studies evaluating reduced food intake, weight management and exercise regimens are reported to improve cancer prognosis and survival (Calle et al, 2003; Courneya et al, 2025; de Cabo and Mattson, 2019; Sjoholm et al, 2023). Furthermore, due to the recent availability of effective anorexigenic and anti-obesogenic pharmacologic interventions, there is an emergence of complementary interest in their potential anticancer benefits (Wang et al, 2024).

Metformin, an orally bioavailable biguanide drug, is one of the most frequently prescribed insulin-sensitizing drugs worldwide (Bailey and Turner, 1996), and has known anorexigenic and anti-obesogenic effects (Lee and Morley, 1998; Yerevanian and Soukas, 2019). Metformin has shown a favorable safety profile in cancer clinical trials (Lord and Harris, 2023); however, its effects on metabolic regulation and weight management are not well-understood in this setting. A recent study (Xiao et al, 2024) reported that metformin promotes appetite suppression and weight control through the production of N-lactoyl phenylalanine (Lac-Phe), a metabolite generated in gut cells through mass action of the enzyme carnosine dipeptidase II (CNDP2) on lactate (Li et al, 2022). This study (Xiao et al, 2024) also found that mean circulating Lac-Phe levels were elevated in post-metformin treatment vs. baseline in Type 2 diabetic (T2D) non-oncologic patients (Abbasi et al, 2004). Furthermore, normalized Lac-Phe levels were elevated in participants on metformin vs. those not on metformin from the Multi-Ethnic Study of Atherosclerosis (MESA) trial (Bild et al, 2002). Mediation analysis of data from the MESA studies also identified a potential role for elevated Lac-Phe in metformin-induced weight loss (Xiao et al, 2024). Lac-Phe was initially reported to be induced by exercise and to underlie its anorexigenic and anti-obesogenic effects (Li et al, 2022). Thus, these collective findings raise the intriguing possibility that metformin use in cancer patients could recapitulate the metabolic benefits of exercise and weight management by elevating Lac-Phe levels, thereby potentially improving cancer outcomes.

Metabolic dysregulation is a hallmark of cancers. To determine whether the link between metformin and elevated Lac-Phe can be

[1]Department of Medicine, Division of Medical Oncology, University of Miami Miller School of Medicine, Miami, FL, USA. [2]Sylvester Comprehensive Cancer Center, Miami, FL, USA. [3]Department of Pathology & Laboratory Medicine, University of Miami Miller School of Medicine, Miami, FL, USA. [4]College of Arts and Sciences, University of Miami, Coral Gables, FL, USA. [5]Department of Radiation Oncology, Division of Biology, University of Miami Miller School of Medicine, Miami, FL, USA. [6]Sheila and David Fuente Graduate Program in Cancer Biology, University of Miami Miller School of Medicine, Miami, FL, USA. [7]Department of Molecular and Cellular Pharmacology, University of Miami Miller School of Medicine, Miami, FL, USA. [8]Miami VA GRECC, Miami, FL, USA. ✉E-mail: mxb2305@med.miami.edu; dbl68@med.miami.edu; prai@med.miami.edu

generalized to the cancer context, we measured serum levels of Lac-Phe in prostate cancer patients. We had several reasons for focusing on prostate cancer patients. Obesity and metabolic dysfunction are known drivers of treatment failure and lethal progression in prostate cancer (Saha et al, 2023). Moreover, standard-of-care hormone therapies can induce significant weight gain and metabolic syndrome, thus increasing the risk of fatal treatment-associated cardiovascular events (Braga-Basaria et al, 2006; Timilshina et al, 2012; Troeschel et al, 2020). In this study, we profiled samples from a prospective metformin-based trial, BIMET-1 (NCT02614859) (Bilusic et al, 2022), where all enrolled patients were androgen deprivation therapy (ADT)-naïve, either overweight or obese (but non-diabetic), and were diagnosed with biologically-recurrent hormone-sensitive prostate cancer at high risk for metastatic progression (Bilusic et al, 2022). The primary endpoint of this trial was to assess whether the anti-androgen, bicalutamide, combined with metformin, would delay the onset of metastases while preserving quality of life. The trial design incorporated metformin monotherapy as a key secondary endpoint for the first cycle (8 weeks/D56). Patients received metformin doses of 1000 mg BID. Although the primary endpoint of enhancing bicalutamide efficacy was not reached, metformin monotherapy nevertheless prevented PSA rise in about 40% of treated high-risk patients, who were classified as responders (R) (Bilusic et al, 2022). Thus, Lac-Phe profiling in this patient cohort enables assessment of its correlation with anticancer response and comparative effects on weight management from a controlled metformin regimen pre- and post-assignment to anti-androgen therapy. The trial arms and patient cohort characteristics, as well as the number of samples profiled in our study, are summarized in Figs. 1A and EV1A.

We profiled all available (12) BIMET-1 patient samples out of the original 29 patients to determine whether Lac-Phe levels associated with metformin anticancer response and/or improved weight management following assignment to bicalutamide. To complement this analysis and assess the ability of metformin treatment to elevate Lac-Phe in an expanded patient cohort, we performed blinded profiling of an additional 25 prostate cancer patients across the high-risk/lethal disease spectrum (metastatic hormone-sensitive, biochemically-recurrent or metastatic castration-resistant), some of whom were taking metformin (either 500 mg BID or 1000 mg BID) for metabolic dysfunction. While Lac-Phe levels did not prognosticate anticancer response to metformin in the BIMET-1 patients, our study shows for the first time that metformin use consistently and robustly elevates Lac-Phe levels in prostate cancer patients, regardless of age, disease stage, BMI, or other treatments, including anti-androgen therapies. Moreover, the metformin-induced Lac-Phe concentrations in these patients were comparable with those reported for strenuous exercise (Li et al, 2022).

## Results and discussion

### Metformin treatment significantly elevates Lac-Phe levels in prostate cancer patients, independent of disease stage, BMI, and hormone therapy status

Targeted metabolic profiling for lactate and Lac-Phe, using ultra-high-performance liquid chromatography and tandem mass spectrometry (UHPLC-MS/MS), was performed on all available retrospective serum samples from both BIMET-1 trial arms across all timepoints (D1, D56, D225) (Fig. 1A) as well as from prostate cancer patients enrolled in an umbrella biomarker protocol. Absolute values of metabolites were calculated against an internal standard. The Lac-Phe signal in serum samples was confirmed by matching both the retention time and the mass spectrometry spectrum of an authentic chemical standard (Appendix Fig. S1). Consistent with the Xiao et al study (Xiao et al, 2024), metformin treatment did not induce significant changes in mean lactate concentrations (Figs. 1B and EV1B). Importantly, despite lower baseline lactate levels in Arm B vs. Arm A patients, metformin treatment significantly increased mean Lac-Phe levels from baseline in all Arm B patient samples (Fig. 1C). By contrast, Lac-Phe concentrations were not significantly altered in baseline Arm A vs. bicalutamide-treated Arm A patients (Fig. 1C). However, unlike Arm A patients who were on bicalutamide alone, Lac-Phe was significantly elevated in Arm B patients who underwent six months of bicalutamide co-treatment with metformin (Fig. 1C). To determine whether Lac-Phe levels could predict patient anticancer response to metformin, we assessed changes in Lac-Phe levels before and after treatment in individual BIMET-1 patients, classified as R (metformin-responsive) or NR (metformin non-responsive) based on changes in their PSA levels (Fig. EV1A). However, changes in Lac-Phe levels did not consistently distinguish R vs NR patients in either the metformin monotherapy or bicalutamide co-treatment cohorts and, thus, were not predictive of metformin's anticancer efficacy in this small patient cohort (Fig. 1D,E).

The BIMET-1 cohort had the advantage of a homogenous study population where all the patients were overweight or obese, at the hormone-naïve castration-sensitive disease stage, and were on a controlled metformin dose intended for maximal oncologic benefit. However, given how commonly metformin is prescribed for metabolic dysfunction, we wanted to determine whether its benefit in elevating Lac-Phe would extend to a broader prostate cancer population, particularly in castration-resistant prostate cancer (CRPC) patients, who undergo sustained androgen blockade to manage their disease. Therefore, we conducted Lac-Phe measurements in sera from an additional 25 patients across the advanced prostate cancer spectrum, who were enrolled in an umbrella biomarker specimen collection protocol at our cancer center. Patient characteristics from this non-trial cohort at the time of blood collection are summarized in Fig. EV1C. Profiling was performed blinded as to patient treatment status. Following unblinding, 10 out of these 25 patients were found to be on metabolism-modifying interventions with 7 patients on metformin (doses ranging from 500 to 1000 mg BID), indicated in red in Fig. 1F. Subsequent analysis revealed this group had significantly elevated mean Lac-Phe levels compared to the other 15 patients who were not on any metabolism-modifying interventions (Figs. 1F and EV1C). The other three patients showing elevated Lac-Phe were on other metabolism-modifying drugs (insulin, tirzepatide, or semaglutide). Feeding has also been reported to increase post-prandial Lac-Phe levels, independent of metformin treatment, in non-diabetic individuals (Scott et al, 2024). Blood from BIMET-1 patients was collected in a fasting state, thus removing a potential post-prandial contribution to their Lac-Phe levels. Not all the samples in the 25 patients from the biomarker protocol were

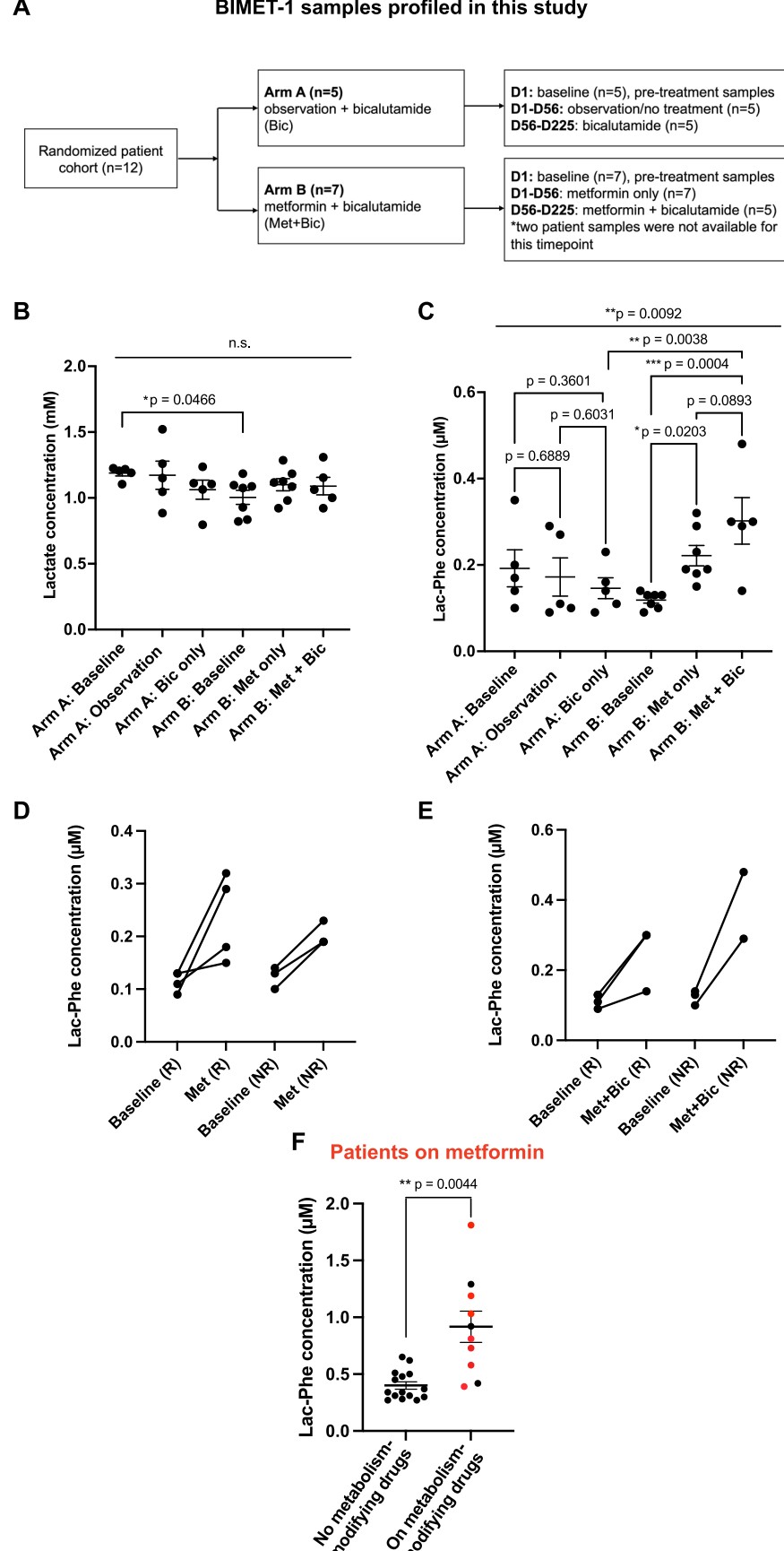

**A** BIMET-1 samples profiled in this study

**B**

**C**

**D**

**E**

**F** Patients on metformin

◄ **Figure 1. Metformin treatment significantly increases Lac-Phe levels in patient sera, independent of disease stage, anti-androgen therapy and metformin-induced anticancer response.**

(A) Schematic of the BIMET-1 trial arms and patient samples profiled in the study. (B) Lactate concentrations (mM) in BIMET-1 patients. Comparison of lactate levels across the indicated Arm A ($n = 5$) and Arm B ($n = 7$) patient sera. Each point represents an individual patient. Data are representative of two technical replicates. Error bars represent SEM. Ordinary one-way ANOVA with Fisher's LSD test. Overall significance and significant pairwise comparisons with their associated $P$ values are noted in the graph. (C) Lac-Phe concentrations (µM) in BIMET-1 patients. Comparison of Lac-Phe levels across the indicated Arm A ($n = 5$) and Arm B ($n = 7$) patient sera. Each point represents an individual patient. Data are representative of two technical replicates. Error bars represent SEM. Ordinary one-way ANOVA with Fisher's LSD test. Overall significance and significant pairwise comparisons with their associated $P$ values are noted in the graph. (D) Trendlines showing changes in Lac-Phe at the D56 (metformin monotherapy) vs. D1 (baseline) timepoints for individual Arm B patients who were classified as metformin responders (R) or non-responders (NR). (E) Trendlines showing changes in Lac-Phe at the D225 (metformin + bicalutamide) vs. D1 timepoints for individual Arm B patients who were classified as metformin responders (R) or non-responders (NR). (F) Lac-Phe concentrations (µM) in prostate cancer patients from the umbrella biomarker protocol at the University of Miami/Sylvester Comprehensive Cancer Center. Comparison of Lac-Phe levels between patients receiving metabolism-modifying drugs ($n = 10$) and those not receiving such agents ($n = 15$). Patients on metformin ($n = 7$) are indicated in red. Each point represents an individual patient. Error bars represent SEM. Unpaired Student's $t$ test with Welch's correction for unequal variance. Source data are available online for this figure.

collected in a fasted state, and this information was not available to us. However, significantly elevated Lac-Phe levels were only observed in the patients on metabolism-modifying interventions, and the variation in Lac-Phe concentrations for patients not on metabolism-modifying drugs was comparatively smaller (Fig. 1F). These observations suggest that post-prandial contributions to Lac-Phe are likely to be minor compared to those from metformin or other metabolism-regulating agents. Overall, our findings in this cohort support the robustness of the metabolic link between metformin use and Lac-Phe in prostate cancer patients across high-risk and advanced disease, including those on hormone therapy. Moreover, our results suggest other anti-obesogenic interventions may also be able to elevate Lac-Phe, positioning it as a potentially unifying metabolism-regulatory node at the interface of glycemic control and adiposity.

Lac-Phe is also induced by strenuous exercise (Li et al, 2022). The recent CHALLENGE trial has demonstrated the significant benefits of structured exercise alone on survival in high-risk colorectal patients (Courneya et al, 2025). Exercise-induced Lac-Phe levels in retrospectively analyzed samples from non-diabetic healthy individuals (Contrepois et al, 2020) fall in the 0.1–0.25 µM range (Li et al, 2022). In our study, metformin-induced Lac-Phe concentrations were in the 0.1–0.35 µM range for the BIMET-1 cohort (Fig. 1C) and in the 0.3-1.7 µM range for the biomarker protocol-enrolled prostate cancer patients (Fig. 1F). Thus, our results warrant further investigation into whether metformin treatment can pharmacologically recapitulate the metabolic benefits of exercise in improving cancer outcomes through induction of Lac-Phe, particularly in those cancer patients who may be unable to exercise.

## Lac-Phe, rather than growth/differentiation factor 15 (GDF-15), is associated with improved weight management in metformin-treated prostate cancer patients

Because of the significant weight gain and increased risk of cardiovascular sequelae associated with standard-of-care hormone therapies for prostate cancer, actionable mechanisms that can promote weight management in these patients are of significant clinical interest. Being overweight (BMI > 25) was a key enrollment criterion for the BIMET-1 trial (Fig. EV1A), and weight was monitored throughout the trial (Bilusic et al, 2022). The

metformin-treated BIMET-1 (Arm B) patients showed significantly better weight management following six months of bicalutamide co-treatment compared to those who did not receive metformin (Arm A), with only one Arm B patient gaining weight over the trial period (Figs. 2A and EV2A). These findings affirm that the benefits of metformin on weight management as well as elevated Lac-Phe are maintained even after co-assignment to anti-androgen therapy (Fig. 1C,F).

While the recent studies on Lac-Phe suggest it plays a major anti-obesogenic role (Li et al, 2022; Xiao et al, 2024), metformin-induced weight management has previously been attributed to its elevation of the cytokine, growth/differentiation factor 15 or GDF-15 (Coll et al, 2020). GDF-15 was the only significantly changed cytokine (out of 105 profiled cytokines) following metformin monotherapy in BIMET-1 patients (Fig. EV2B), and was significantly elevated in all Arm B patients at D56 (Fig. 2B,C). GDF-15 levels did not correlate with Lac-Phe at this timepoint, suggesting these factors are not coordinately influenced by metformin (Fig. EV2C). As all Arm B patients were hormone-naïve and lost weight on metformin alone (D1-D56) (Figs. 2A and EV2A), we evaluated whether the extent of weight loss in this setting correlated with the changes in Lac-Phe and/or GDF-15 levels. While neither Lac-Phe nor GDF-15 correlated significantly with net patient weight loss, the overall trend of patient weight management correlated more strongly with elevated Lac-Phe than with GDF-15 (Fig. 2D,E). Thus, while not conclusive, our results support findings from other studies that report elevated Lac-Phe underlies metformin-induced weight loss and that, despite its increased levels, the GDF-15 axis is likely to be dispensable for metformin-induced weight loss (Klein et al, 2022; Xiao et al, 2024). Overall, our results indicate that Lac-Phe could be an important molecular factor associated with weight management in prostate cancer patients, particularly those at increased risk of suffering metabolically adverse effects from anti-androgen therapy.

Our findings connecting Lac-Phe elevation to metformin treatment and weight management following hormone therapy provide a novel molecular context to the recently published STAMPEDE trial, a major prospective study in prostate cancer patients that found metformin improves weight management and lowers glucose, independent of its anticancer effects (Gillessen et al, 2025). More broadly, with the natural caveat that our study does not include female cancer patients, our results generalize the metabolic link between metformin and Lac-Phe, first observed in

T2D patients and healthy individuals (Li et al, 2022; Xiao et al, 2024), to the oncologic context.

# Methods

### Reagents and tools table

| Reagent/resource | Reference or source | Identifier or catalog number |
|---|---|---|
| **Experimental models** | | |
| Patient serum samples | BIMET-1 trial | NCT02614859 |
| Patient serum samples | Umbrella biomarker specimen collection protocol at University of Miami/Sylvester Comprehensive Cancer Center | IRB approval #20250101 |
| **Recombinant DNA** | | |
| | Not applicable | |
| **Antibodies** | | |
| IRDye 800CW Streptavidin (cytokine array) | R&D Systems | CAT# 926-32230 (RRID not available) |
| **Oligonucleotides and other sequence-based reagents** | Not applicable | |
| **Chemicals, enzymes, and other reagents** | | |
| Sodium L-lactate (98%) | Millipore Sigma, USA | 1614308-200MG |
| N-lactoyl-phenylalanine (>95%) | Cayman Chemical | 37304 |
| Sodium L-lactate ($^{13}C_3$, 98%) | Cambridge Isotope Laboratories | CLM-1579-N-0.1MG |
| LCMS grade methanol | Millipore Sigma, USA | 1.06035.1000 |
| LCMS grade water | Honeywell Research Chemicals, USA | 142632 L |
| LCMS grade Acetonitrile | Millipore Sigma, USA | AX0156 |
| Tributylamine (TBA) | Millipore Sigma, USA | 90781 |
| Glacial acetic acid | Millipore Sigma, USA | AX0074 |
| **Software** | | |
| GraphPad Prism V. 10.3.1 | https://www.graphpad.com/ | RRID:SCR_002798 |
| Agilent Masshunter Quantitative Software 12.1 | Agilent, USA | RRID:SCR_015040 |
| Agilent Masshunter acquisition software 12.1 | Agilent, USA | |
| Image Studio Software | LICORbio™ | RRID:SCR_015795 |
| **Other** | | |
| Agilent 1290 ultrahigh-performance liquid chromatography (UHPLC) | Agilent, USA | SCR_019375 |
| 6495 C QqQ mass spectrometer | Agilent, USA | G6495CA |
| UHPLC Guard | Agilent, USA | Part no. 821725-907 |

| Reagent/resource | Reference or source | Identifier or catalog number |
|---|---|---|
| Agilent Zobrax extend C18 column | Agilent, USA | Part no. 759700902 |
| Vacuum concentrator system | Labconco, USA | 7310020, 7460020, 7393000 |
| Proteome Profiler Human XL Cytokine Array Kit | R&D Systems | ARY022B |
| LI-COR Odyssey M imaging system | LICORbio™ | Odyssey DLx (RRID:SCR_025709) |

## Sex as a biological variable

As the focus of this study is prostate cancer patients, only specimens from males are included.

## Blood collection and weight measurements

All experiments in this study conformed to the principles set forth in the WMA Declaration of Helsinki and the Department of Health and Human Services Belmont Report.

The details of the study design and patient enrollment criteria for the BIMET-1 trial (NCT02614859) have been previously described (Bilusic et al, 2022) and are summarized in Figs. 1A and EV1A. Patients whose PSA remained stable or decreased during the metformin monotherapy cycle were classified as responders (R) whereas those that showed an increase in PSA were classified as non-responders (NR). Patient B-16 was classified as a partial responder as this patient exhibited a rapid decrease in PSA during the first four weeks, followed by a PSA increase. Blood was collected serially from the patients (fasted for at least 12 h) on day 1, day 56, and day 225 in serum separator tubes. Collected blood specimens were kept at room temperature for 30 min, centrifuged and immediately saved at −80 °C for future analysis. The current study represents a subset of BIMET-1 patients whose sera were available for further analyses. The BMI levels for the respective patients were also monitored and recorded at the beginning of each cycle. The samples collected at baseline, after two months (56 days), and at the end of the study (225 days) were compared to evaluate the weight-lowering effect of introducing metformin.

Sera from prostate cancer patients, across all disease stages, enrolled in an umbrella biomarker specimen collection protocol at University of Miami/Sylvester Comprehensive Cancer Center (IRB#20250101) were also profiled for lactate and Lac-Phe levels. Specimen storage and processing were done similarly as the BIMET-1 patient samples. Informed consent was obtained from all participants, and studies were approved by an Institutional Review Board.

## LC-MS measurements and quantitation of serum lactate and Lac-Phe levels

LC-MS profiling was carried out blinded as to patient therapeutic regimens, and data were unblinded into specific groups during final analyses.

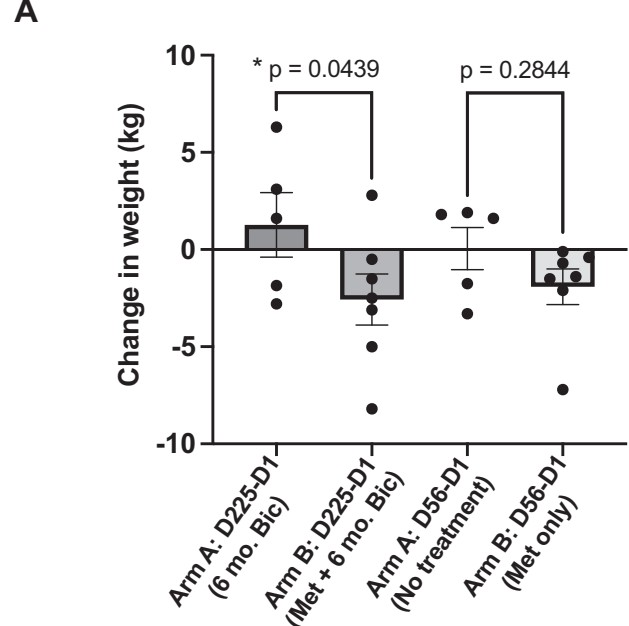

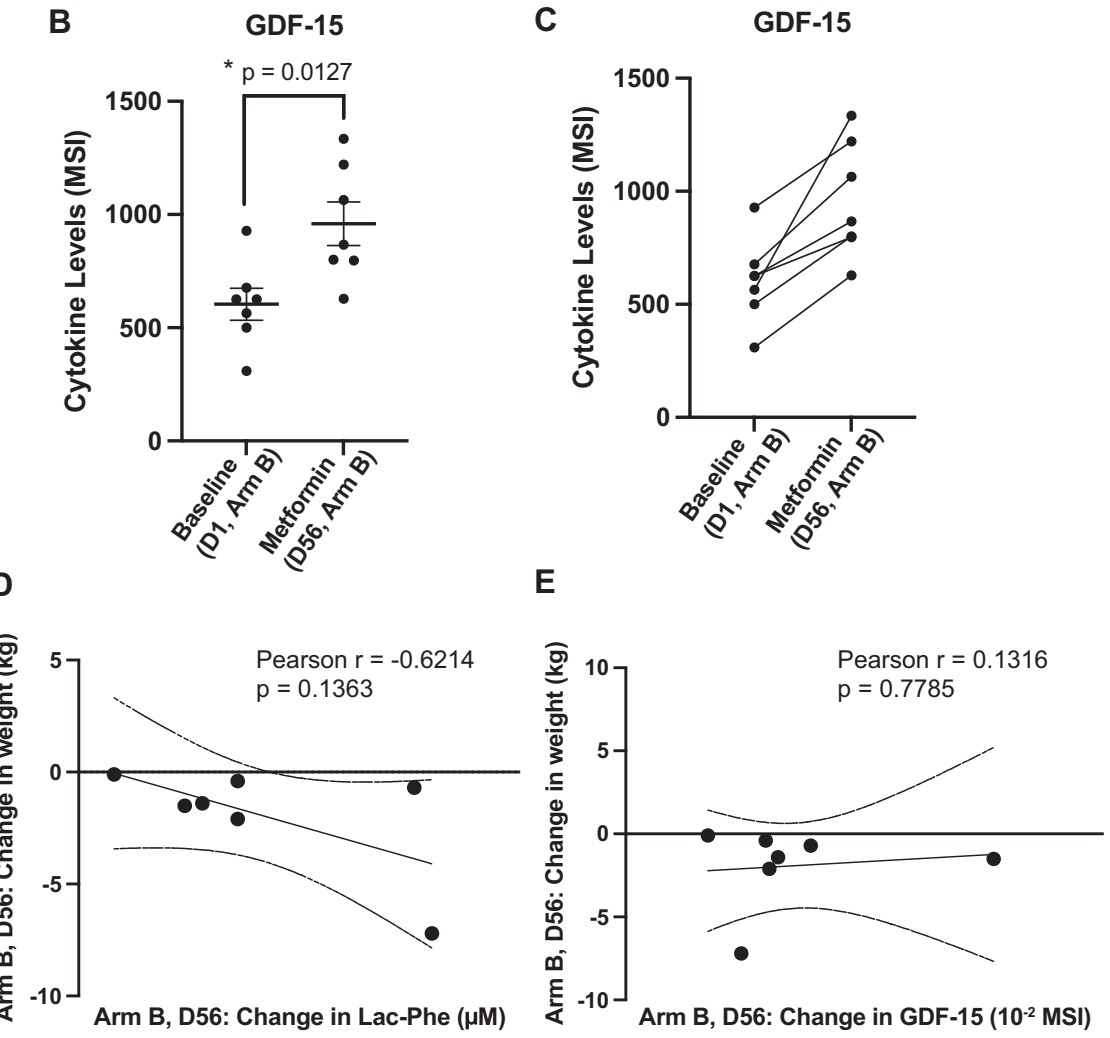

**Figure 2. Metformin-associated weight management in prostate cancer patients shows a stronger association with Lac-Phe than with GDF-15.**

(**A**) Change in weight in Arm A ($n = 5$) vs. Arm B ($n = 7$) patients at the indicated trial timepoints. Ordinary one-way ANOVA with Fisher's LSD test. (**B**) Mean signal intensity (MSI) of GDF-15 levels at baseline (D1) and metformin monotherapy (D56) in Arm B ($n = 7$) patients. Each point represents an individual patient. Values are the averages of two technical replicates. Error bars represent SEM. Unpaired Student's $t$ test with Welch's correction for unequal variance. (**C**) Trendlines showing the differences in GDF-15 levels in individual Arm B patients at the D1 vs. D56 timepoints. (**D**) Correlation between the change in Lac-Phe levels vs. the change in weight for Arm B patients at the D56 timepoint (following metformin monotherapy). The linear regression line and 95% confidence intervals are shown, with Pearson correlation coefficient, $r$, and $P$ value of the correlation noted above the graph. (**E**) Correlation between the change in weight vs. the change in GDF-15 levels (MSI × $10^{-2}$) for Arm B patients, at the D56 timepoint. The linear regression line and 95% confidence intervals are shown, with Pearson correlation coefficient, $r$, and $P$ value of the correlation noted above the graph. Source data are available online for this figure.

A stock solution of sodium L-lactate ($^{13}C_3$, 98%, Cambridge Isotope Laboratories) was prepared at a concentration of 0.05 mg/mL by dissolving in LC-MS grade water (Honeywell Research Chemicals, USA) and used as an internal standard for both lactate and Lac-Phe quantification. Then, 15 µL of the internal standard was added to 50 µL of patient serum. Next, 260 µL of ice-cold methanol was added to 65 µL of the samples (80% v/v methanol). Samples were then incubated on ice for 10 min and centrifuged at 14,000 × $g$ for 10 min at 4 °C. The supernatant was carefully collected into new 1.5-mL tubes, and the solvent was evaporated using a vacuum concentrator system (Labconco, USA). Finally, the dried samples were reconstituted in LC-MS grade water (Honeywell Research Chemicals, USA) for reverse-phase chromatography.

An Agilent 1290 ultrahigh-performance liquid chromatography (UHPLC) and 6495 C QqQ mass spectrometer were employed to quantify serum lactate and Lac-Phe levels. Agilent MassHunter acquisition software 12.1 was used to acquire data. Patient samples were run twice, once to acquire relative Lac-Phe levels and then a second replicate was used to acquire the absolute concentration reported here. The trends between both sets of data were identical across profiled samples. The following method parameters were used for reverse-phase chromatography: 2 µL injection volume, 0.250 mL/min flow, UHPLC Guard (P.N.821725-907) and Agilent Zobrax extend C18 column (P.N. 759700902), 35.0 °C column temperature, Solvent A (LCMS grade water with 3% LCMS grade methanol, 10 mM tributylamine (TBA), and 15 mM glacial acetic acid) and solvent C (LCMS grade methanol with 10 mM tributylamine (TBA), and 15 mM glacial acetic acid) for chromatographic separation, and solvent D (100% LCMS grade acetonitrile) for column wash. The following gradient was used: 100% A with 0.250 mL/min flow at 0 to 2.5 min, 80% A and 20% C with 0.250 mL/min flow at 7.5 min, 55% A and 45% C with 0.250 mL/min flow at 13 min, 1% A and 99% C with 0.250 mL/min flow at 20 min, 1% A and 99% C with 0.250 mL/min flow at 24 min, 1% A and 99% D with 0.250 mL/min flow at 24.05 min, 1% A and 99% D with 0.250 mL/min flow at 27 min, 1% A and 99% D with 0.800 mL/min flow at 27.5 min, 1% A and 99% D with 0.800 mL/min flow at 31.35 min, 1% A and 99% D with 0.600 mL/min flow at 31.5 min, 100% A with 0.400 mL/min flow at 32.25 min, 100% A with 0.400 mL/min flow at 39.9 min, 100% A with 0.250 mL/min flow at 40 min. The QqQ was run in multiple reaction mode (MRM) with the following parameters: negative electrospray ionization, 25 ms dwell time, 150 °C gas temperature with 13 L/min flow, Nebulizer at 45 psi, 325 °C of Sheath Gas Temperature with 12 L/min flow, 2000 V of capillary, and 500 V of Nozzle voltage.

The absolute levels of lactate and Lac-Phe in each sample were quantified using a calibration curve with Agilent Quantitative Software 12.1. The calibration curves were prepared using freshly-made standard solutions with Sodium L-lactate (98%) from Millipore Sigma, USA (1.56, 0.78, 0.39 mM dissolved in LC-MS grade water) and N-lactoyl-phenylalanine (>95%) from Cayman Chemical, USA (10 µM, 5 µM, 2.5 µM, 1.25 µM, 0.625 µM, 0.3125 µM, 0.1563 µM, 0.078 µM, 0.0391 µM dissolved in LC-MS grade water), for lactate and Lac-Phe, respectively.

## Cytokine profiling

Cytokine profiling was carried out using the Proteome Profiler Human XL Cytokine Array Kit (R&D Systems, ARY022B) on D1 or D56 sera from Arm B patients. IRDye 800CW Streptavidin (CAT 926-32230) was used as the secondary antibody at 1:10,000 dilution. The LI-COR (Odyssey M) imaging system was used to detect signal intensity, and Image Studio Software was used for quantification. Each sample was run in two technical replicates, and the two values were averaged to represent the plotted cytokine values. Pre- and post-treatment levels were analyzed on GraphPad Prism (V.10.3.1) to determine statistically significant ($P < 0.05$, unpaired Student's $t$ test, with Welch's correction) altered cytokines.

## Statistical methodology

The patients on the BIMET-1 trial were randomized 1:2 to Arm A (observation × 2 cycles, followed by bicalutamide 50 mg PO daily × 6 cycles) or Arm B (metformin 1000 mg twice daily × 2 cycles, followed by metformin 1000 mg twice daily plus bicalutamide 50 mg PO daily x 6 cycles) and stratified according to prior definite therapy (radiation vs. surgery). Blood samples were taken at prespecified time points, after 8 weeks, and at the end of the trial (32 weeks). All the statistical analyses were performed in GraphPad Prism (V. 10.3.1). All error bars represent standard error of the mean (SEM). For comparison across two groups, an unpaired Student's $t$ test with Welch's correction was employed. For comparison across more than two groups, an ordinary one-way ANOVA test was performed with a Fisher's LSD test to determine pairwise significance. For correlation analyses between two variables, the Pearson coefficient, $r$, was used to determine the strength of their linear correlation. $P$ value < 0.05 was considered statistically significant and expressed with *; *$P < 0.05$, **$P < 0.01$, ***$P < 0.001$, ****$P < 0.0001$.

## The paper explained

### Problem

Cancer remains a leading cause of global mortality, and excess body weight is a well-established risk factor associated with poorer cancer prognosis and treatment outcomes. In prostate cancer specifically, obesity and metabolic dysfunction contribute to treatment resistance and lethal disease progression, while standard-of-care hormonal therapies frequently exacerbate weight gain and cardiometabolic risk. Although lifestyle interventions such as exercise and dietary restriction improve outcomes, their implementation and durability in cancer patients are often limited. There is therefore a clinical need for safe, pharmacologic strategies that improve metabolic health and weight control in cancer patients without compromising oncologic care.

### Results

Metformin, a widely used insulin-sensitizing drug with known anorexigenic and anti-obesogenic effects, has been shown in non-cancer populations to elevate the exercise-induced metabolite, N-lactoyl phenylalanine (Lac-Phe), which mediates appetite suppression and weight control. In this study, we examined whether this metabolic pathway extends to cancer. Using serum samples from the BIMET-1 prospective trial as well as an independent cohort of prostate cancer patients across disease stages, we found that metformin robustly and consistently elevated circulating Lac-Phe levels. This effect was observed irrespective of disease stage, initial BMI, or concurrent anti-androgen therapy, and the magnitude of Lac-Phe elevation was comparable to the concentrations reported after strenuous exercise. While Lac-Phe levels did not predict anticancer response to metformin, metformin use was associated with improved weight management, particularly in patients receiving hormonal therapies.

### Impact

These findings establish, for the first time, that metformin-induced elevation of Lac-Phe occurs in prostate cancer patients, extending a previously described metabolic mechanism from non-oncologic populations into the cancer setting. By recapitulating key metabolic benefits of exercise, metformin may help mitigate therapy-induced weight gain and metabolic dysfunction in prostate cancer patients, especially those who are unable to exercise. It remains to be seen if this pathway also operates in other cancer patients, particularly female patients. Regardless, our study highlights Lac-Phe as a potential molecular mediator of metabolic benefit from metformin use in cancer patients, independent of its anticancer efficacy, with implications for improving long-term outcomes and quality of life in cancer care.

## Graphics

The synopsis image was created with BioRender.com.

## Data availability

This study includes no data deposited in external repositories.

The source data of this paper are collected in the following database record: biostudies:S-SCDT-10_1038-S44321-026-00408-6.

## Peer review information

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

## Acknowledgements

This work was supported by funding through R01CA254100 (PR), Women's Cancer Association (WCA) cancer research award (PR), and Sylvester Comprehensive Cancer Center research development funds (PR, MB, JS). Work in the Lombard lab is supported by R01CA253986 and R33AG077856, as well as the DoD (ME200030), Florida Department of Health (24B12), Melanoma Research Alliance (1434401), and the Miami VA Healthcare System GRECC (I01BX006593). We thank Sergio Rodriguez Jr., Helen Khattoura, and Melinda Boone from the Sylvester Biospecimen Shared Resource (BSSR) for assistance with curating patient samples. We thank Dr. Scott Welford for helpful discussions. Research reported in this publication, including metabolomics, was supported by the National Cancer Institute of the National Institutes of Health under Award Number P30CA240139. The content is solely the responsibility of the authors and does not necessarily represent the official views of the National Institutes of Health.

## Author contributions

**Marijo Bilusic**: Conceptualization; Formal analysis; Supervision; Investigation; Writing—original draft; Project administration; Writing—review and editing. **Durga Prasad Gannamedi**: Resources; Software; Formal analysis; Methodology; Writing—original draft. **Bhipasha Challu**: Validation; Investigation; Writing—original draft; Writing—review and editing. **Shomita Ferdous**: Data curation; Formal analysis; Validation; Writing—original draft; Writing—review and editing. **Beatriz Mateo-Victoriano**: Resources; Investigation; Methodology; Writing—review and editing. **Sheela Pokharel**: Resources; Investigation; Methodology; Writing—review and editing. **Defne Bayik**: Conceptualization; Investigation; Writing—original draft; Writing—review and editing. **Janaki Sharma**: Resources; Formal analysis; Investigation; Writing—original draft; Writing—review and editing. **David B Lombard**: Conceptualization; Resources; Data curation; Formal analysis; Supervision; Methodology; Writing—original draft; Writing—review and editing. **Priyamvada Rai**: Conceptualization; Data curation; Formal analysis; Supervision; Funding acquisition; Validation; Investigation; Writing—original draft; Project administration; Writing—review and editing.

Source data underlying figure panels in this paper may have individual authorship assigned. Where available, figure panel/source data authorship is listed in the following database record: biostudies:S-SCDT-10_1038-S44321-026-00408-6.

## Disclosure and competing interests statement

M Bilusic has received research support via his institution from Surface Oncology; Merck Sharp & Dohme; Eli Lilly & Co; Hookipa Pharma; Marengo Therapeutics and Bicycle Therapeutics. The remaining authors declare no competing interests.

# Expanded View Figures

**Figure EV1.   Information for patient cohorts from the BIMET-1 trial and from the University of Miami/Sylvester Comprehensive Cancer Center's umbrella biomarker protocol who were profiled in this study.**

(A) Age, PSA levels and BMI for the patients ($n = 5$) in Arm A and patients ($n = 7$) in Arm B from the BIMET-1 trial. The patients in Arm B are further subdivided into responders, R (in red), or non-responders, NR. Positive response was defined as a decline in serum PSA. *One patient was categorized as a partial responder due to a rapid decrease in PSA for the first four weeks of metformin monotherapy but then followed by an increasing PSA. **Patient sera samples that were unavailable at the end of D225. (B) Trendlines showing changes in lactate at the D56 timepoint vs. D1 for individual Arm B patients from the BIMET-1 trial who were classified as metformin responders (R) or non-responders (NR). (C) Disease stage, age, BMI, PSA levels, and hormone therapy status for prostate cancer patients enrolled through an umbrella biomarker specimen collection protocol and profiled in this study for Lac-Phe levels in Fig. 1F. Source data are available online for this figure.

**A**

| Patient ID | Days of blood collection | Age | Gleason | Baseline PSA | PSA (56d) | PSA (225d) | Arm | Met Response (PSA decline 56d) | BMI (baseline) | BMI (56d) | BMI (225d) |
|---|---|---|---|---|---|---|---|---|---|---|---|
| B-2 | 1, 56, 225 | 67 | 4+3=7 | 3.72 | **3.53** | 0.02 | metformin | **R** | **28.3** | **28.1** | **28.1** |
| B-9 | 1, 56, 225 | 64 | 4+4=8 | 5.41 | **5.2** | 0.36 | metformin | **R** | **29.9** | **29.1** | **29.2** |
| B-12 | 1, 56, 225 | 53 | 4+5=9 | 2.09 | **1.88** | 0.7 | metformin | **R** | **27.1** | **24.2** | **25.7** |
| B-16* | 1, 56 | 72 | 4+3=7 | 0.88 | **0.93** | 0.3 | metformin | **R*** | **28.4** | **28.3** | **25.8**** |
| B-3 | 1, 56, 225 | 74 | 3+5=8 | 16.26 | 20.22 | 2.14 | metformin | NR | 25.6 | 25.4 | 26.3 |
| B-14 | 1, 56, 225 | 58 | 4+5=9 | 4.07 | 5.1 | 0.14 | metformin | NR | 30.4 | 29.8 | 29.5 |
| B-15 | 1, 56, 225 | 73 | 4+3=7 | 10.76 | 13.1 | 0.66 | metformin | NR | 25.9 | 25.4 | 25.1** |
| A-1 | 1, 56, 225 | 63 | 4+3=7 | 1.18 | 1.53 | 0.06 | observation | NR | 33.1 | 33.6 | 35 |
| A-6 | 1, 56, 225 | 68 | 3+4=7 | 10.99 | 16.23 | 2.8 | observation | NR | 29.4 | 30.1 | 30 |
| A-8 | 1, 56, 225 | 60 | 5+4=9 | 7.48 | 8.27 | 1.49 | observation | NR | 33.8 | 33.2 | 33.1 |
| A-10 | 1, 56, 225 | 56 | 3+4=7 | 22.09 | 27.05 | 2.26 | observation | NR | 34 | 32.9 | 33 |
| A-13 | 1, 56, 225 | 65 | 4+4=8 | 5.07 | 7.18 | 0.3 | observation | NR | 37.2 | 37.3 | 38.2 |

**\*partial responder - initial rapid decrease in PSA during the first four weeks, followed by an increase**
**\*\*sera samples not available for profiling at this timepoint**

**B**

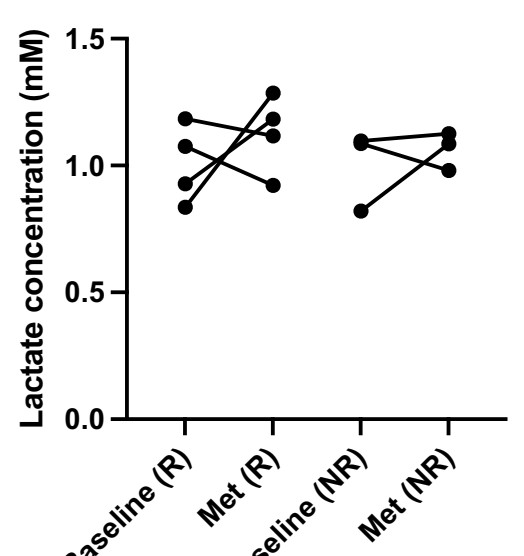

**C**

| Patient Characteristics (n=25) | No metabolism-modifying drugs (15 patients) | On metabolism-modifying drugs (10 patients) |
|---|---|---|
| **Disease stage (% total patients)** | mHSPC/BCR: 53.3% mCRPC: 46.7% | mHSPC/BCR: 60% mCRPC: 40% |
| **Age (mean +/- SEM)** | 70.7+/-1.7 | 71+/-2.5 |
| **BMI (mean +/-SEM)** | 25.6+/-1.1 | 31.7+/-1.5 |
| **PSA (mean +/-SEM)** | 59.05 +/- 31.1 | 2.1 +/- 0.87 |
| **On ADT or ARSi** | 12/15 (80%) | 8/10 (80%) |
| **% of patients on metabolism-modifying drugs** | n/a | Metformin: 70% Tirzepatide: 10% Insulin: 10% Semaglutide: 10% |

**A**

| Patient ID | WEIGHT (baseline, kg) | WEIGHT (56d, kg) | WEIGHT (225d, kg) |
|---|---|---|---|
| B-2 | 91.2 | 90.5 | 90.7 |
| B-9 | 78.6 | 77.2 | 77.1 |
| B-12 | 77.2 | 70 | 72.2 |
| B-16 | 88.9 | 88.8 | 80.7 |
| B-3 | 80.1 | 79.7 | 82.9 |
| B-14 | 98.4 | 96.3 | 95.3 |
| B-15 | 79 | 77.5 | 76.5 |
| A-1 | 106.3 | 108.1 | 112.6 |
| A-6 | 82.9 | 84.8 | 84.5 |
| A-8 | 101.95 | 100.2 | 100.1 |
| A-10 | 103.5 | 100.2 | 100.7 |
| A-13 | 114.2 | 115.8 | 117.3 |

**B**

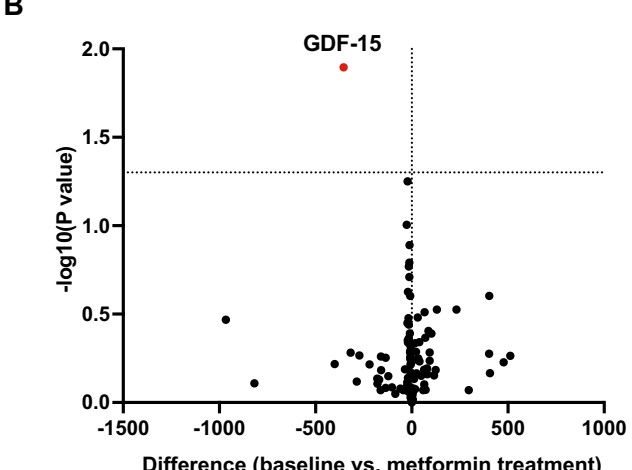

**C**

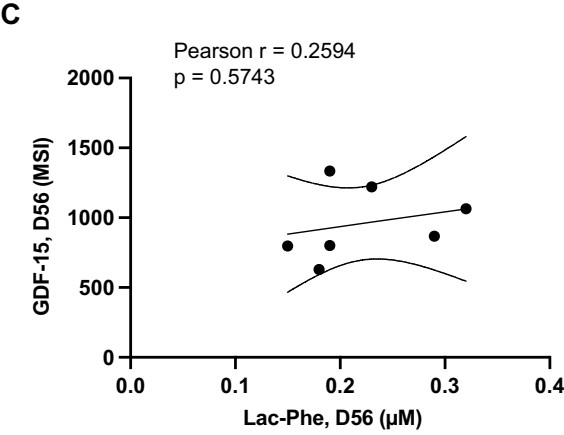

**Figure EV2. GDF-15 is elevated by metformin treatment but does not significantly correlate with Lac-Phe levels.**

(A) The weight of all patients ($n = 12$) at baseline (before treatment or observation), 2 months post-metformin treatment or observation (D56) and at the end of the trial (D225, including 6 months of metformin + bicalutamide, Met + Bic). (B) Volcano plot of significant changes in cytokine levels (mean signal intensity, MSI) following metformin monotherapy (D56), measured in Arm B patient sera. D56 values were compared to sera values at D1 and statistical significance determined via an unpaired Student's *t* test with Welch's correction for unequal variance. (C) Correlation between Lac-Phe levels (μM) vs. GDF-15 levels (MSI) at the D56 (metformin monotherapy) timepoint for Arm B patients. The linear regression line and 95% confidence intervals are shown, with Pearson correlation coefficient, *r*, and *P* value of the correlation noted above the graph. Source data are available online for this figure.

