## [Peer Review File · EMBO Molecular Medicine]

The anti-obesogenic metabolite, Lac-Phe, is elevated by metformin treatment in prostate cancer patients

Marijo Bilusic, Durga Gannamedi, Bhipasha Challu, Shomita Ferdous, Beatriz Mateo-Victoriano, Sheela Pokharel, Defne Bayik, Janaki Sharma, David Lombard, and Priyamvada Rai

Corresponding authors: Priyamvada Rai (prai@med.miami.edu), Marijo Bilusic (mxb2305@med.miami.edu), David Lombard (dbl68@med.miami.edu)

Review Timeline:

Submission Date:	1st Aug 25
Editorial Decision:	3rd Sep 25
Appeal:	12th Sep 25
Editorial Decision:	3rd Oct 25
Revision Received:	16th Dec 25
Editorial Decision:	6th Feb 26
Revision Received:	26th Feb 26
Accepted:	3rd Mar 26

Editor: Lise Roth

Transaction Report:

3rd Sep 2025

Decision on your manuscript EMM-2025-22366

Dear Dr. Rai,

Thank you for submitting your manuscript to EMBO Molecular Medicine. Please accept our apologies for the delay in responding, which is due to referees' and editorial staff's annual leave. We have now received the three referee reports. While referee #1 is supportive of the study overall, referees #2 and #3 are more negative, raising major concerns relating to the limited novelty, small sample size and preliminary nature of the findings.

Due to the nature of these concerns, unfortunately I have little choice but to return the manuscript to you at this time with the decision that we cannot offer to publish it.

I am very sorry to disappoint you this time, but I hope that the referees' comments will be helpful for your continued work in this area.

With kind regards,

Lise Roth

Lise Roth
Senior Editor
EMBO Molecular Medicine

***** Reviewer's comments *****

Referee #1 (Comments on Novelty/Model System for Author):

This is a technically well done study
The novelty is considered medium because this is a replication and extension of the original Xiao et al Nature Metabolism paper, now in a new cohort of prostate cancer patients
The medical impact is high because it is important to understand the effect of metformin on these patients, and how general the metformin/Lac-Phe connection actually is

Referee #1 (Remarks for Author):

This is a well done study that shows metformin-inducible increases in both Lac-Phe and GDF15 in a small cohort of prostate cancer patients. The technical mass spec aspects of this study are well done. The patient cohort is well characterized. The authors also extend the analysis to GDF15, which is good and important because these two molecules seem to both be inducible by similar stimuli including metformin. In my opinion this is suitable for publication.

Referee #2 (Remarks for Author):

This observational study shows that the metabolite Lac-Phe is elevated in cancer patients treated with metformin and the increase correlates positively with weight loss. However, elevation of circulating Lac-Phe by metformin treatment and exercise has previously been reported by others (ref 13, 14). The findings from this study are largely confirmatory. Furthermore, the small sample size and the lack of functional investigation limit the establishment of a cause-effect relationship between changes in Lac-Phe and weight loss in this cancer cohort.

Major points:

- (1) The major issue of this study is the small sample size and large interpersonal variations. Among 12 patients, two serum samples are not available.
- (2) while the body weight changes by metformin treatment is rather mild with large variations (only a small portion of patients shows a modest weight loss from Figure 2A), increases in Lac-Phe are detected in almost all the patients. In this connection, the data presented in panel A, C and D of figure 2 appear to be contradictory.
- (3) In Figure 3E, the lack of significant correlation between the changes in body weight and GDF15 might be due to the small sample size. The data can not be used to exclude the involvement of GDF15 in metformin-mediated weight reduction;
- (5) Another limitation of this study is that only male patients are included. It remains unclear whether the observed effect of

metformin on Lac-Phe also occurs in females.

(6) As this study only shows the clinical association between changes in Lac-Phe and body weight, it does not support the conclusion that Lac-Phe protects against treatment-induced weight gain (last sentence of abstract).

Referee #3 (Comments on Novelty/Model System for Author):

The sample size is too limited to support definitive conclusions, and the authors overinterpret the results

Referee #3 (Remarks for Author):

The article by Balusic et al. presents the results of an ancillary analysis of the BIMET-1 clinical trial, a randomized phase 2 study investigating the effects of combining metformin and bicalutamide on prostate cancer recurrence in obese patients. The focus of this ancillary study is N-lactoyl phenylalanine (Lac-Phe), a gut-derived metabolite previously reported to be induced by exercise. The authors seek to determine whether Lac-Phe levels are elevated in patients and whether it can mimic the effects of physical activity. The sole novel finding of this manuscript is the observed increase in serum Lac-Phe concentrations in patients treated with metformin. However, the effects of metformin on weight management have already been documented in the STAMPEDE trial, and there is no significant correlation between weight loss and Lac-Phe concentration, suggesting that these parameters are not linked.

Major Points:

- The authors should include a schematic overview of the clinical trial design.
- While the original BIMET-1 trial enrolled 29 patients, this ancillary study includes only 12 patients. The authors need to address this discrepancy.
- In Figure 2D, the conclusion does not align with the data; the correlation between weight loss and Lac-Phe concentration is not statistically significant.
- Overall, the study's small sample size limits the validity of its conclusions, and the authors overinterpret the findings.

Conclusion: This study is overly preliminary and does not provide substantial new evidence regarding the impact of metformin in patients with prostate cancer.

As a service to authors, EMBO provides authors with the possibility to transfer a manuscript that one journal cannot offer to publish to another EMBO publication. The full manuscript and if applicable, reviewers reports are automatically sent to the receiving journal to allow for fast handling and a prompt decision on your manuscript. For more details of this service, and to transfer your manuscript to another EMBO title please click on Link Not Available

Email 29th September 2026

Dear Dr. Roth

I hope my email finds you well.

I'm reaching out to see if we will be allowed to submit a revised version of our manuscript?

Since our appeal, we acquired and analyzed 22 prostate cancer patient samples for Lac-Phe levels, blinded as to their metformin status. Comparison of the seven patients who are on metformin vs. those who are not showed the former have higher serum Lac-Phe levels, supporting our primary assertion.

Thanks very much,
Priya

3rd Oct 2025

Dear Prof. Rai, Dear Priya,

Thank you for your email asking us to reconsider our decision regarding your manuscript, and for providing us with a detailed appeal letter. I have read it carefully and discussed it with my colleagues here. I have also consulted referee #3 on your appeal, including your most recent email regarding the addition of 22 prostate cancer patient samples.

The referee agreed to reconsider a revised version that would include these new samples. After further discussion within the team, we concluded that, provided you address the referees' concerns as indicated in your appeal letter and add the new samples, you may submit a revised version of your manuscript.

Please note that the manuscript will be re-reviewed and that we cannot guarantee at this stage that the eventual outcome will be favorable. We would thus encourage you to adequately address all concerns raised by the referees. Acceptance or rejection of the manuscript will depend on the completeness of your responses included in the next, final version of the manuscript. For this reason, and to save you from any frustrations in the end, I would strongly advise against returning an incomplete revision. Please attach a covering letter giving details of the way in which you have handled each of the points raised by the referees.

We are expecting your revised manuscript within three months, if you anticipate any delay, please contact us.

We require:

- 1) A .docx formatted version of the manuscript text (including legends for main figures, EV figures and tables). Please make sure that the changes are highlighted to be clearly visible.
- 2) Individual production quality figure files as .eps, .tif, .jpg (one file per figure). For guidance, download the 'Figure Guide PDF' (<https://www.embopress.org/page/journal/17574684/authorguide#figureformat>).
- 3) At EMBO Press we ask authors to provide source data for the main figures. Our source data coordinator will contact you to discuss which figure panels we would need source data for and will also provide you with helpful tips on how to upload and organize the files.
- 4) A .docx formatted letter INCLUDING the reviewers' reports and your detailed point-by-point responses to their comments. As part of the EMBO Press transparent editorial process, the point-by-point response is part of the Review Process File (RPF), which will be published alongside your paper.
- 5) A complete author checklist, which you can download from our author guidelines (<https://www.embopress.org/page/journal/17574684/authorguide#submissionofrevisions>). Please insert information in the checklist that is also reflected in the manuscript. The completed author checklist will also be part of the RPF.
- 6) All Materials and Methods need to be described in the main text using our 'Structured Methods' format. According to this format, the Methods section includes a Reagents and Tools Table (listing key reagents, experimental models, software and relevant equipment and including their sources and relevant identifiers) followed by a Methods and Protocols section describing the methods, ideally using a step-by-step protocol format. The aim is to facilitate adoption of the methodologies across labs. Please download and fill our Reagents and Tools Table template (.docx), which you can find in our author guidelines: <https://www.embopress.org/page/journal/14693178/authorguide#structuredmethods>. When submitting your revised manuscript, please do not include the Reagents and Tools Table in the Methods section of the manuscript but upload it as a separate file choosing the file type "Reagent Table". An example of a Method paper with Structured Methods can be found here: <https://www.embopress.org/doi/10.15252/msb.20178071>
- 7) It is mandatory to include a 'Data Availability' section after the Materials and Methods. Before submitting your revision, primary datasets produced in this study need to be deposited in an appropriate public database, and the accession numbers and database listed under 'Data Availability'. Please remember to provide a reviewer password if the datasets are not yet public (see <https://www.embopress.org/page/journal/17574684/authorguide#dataavailability>).

8) For data quantification: please specify the name of the statistical test used to generate error bars and P values, the number (n) of independent experiments (specify technical or biological replicates) underlying each data point and the test used to calculate p-values in each figure legend. The figure legends should contain a basic description of n, P and the test applied. Graphs must include a description of the bars and the error bars (s.d., s.e.m.). Please provide exact p values.

12) Author contributions: CRediT has replaced the traditional author contributions section because it offers a systematic machine readable author contributions format that allows for more effective research assessment. Please remove the Authors Contributions from the manuscript and use the free text boxes beneath each contributing author's name in our system to add specific details on the author's contribution. More information is available in our guide to authors.

13) Disclosure statement and competing interests: We updated our journal's competing interests policy in January 2022 and request authors to consider both actual and perceived competing interests. Please review the policy <https://www.embopress.org/competing-interests> and update your competing interests if necessary.

13) Every published paper now includes a 'Synopsis' to further enhance discoverability. Synopses are displayed on the journal webpage and are freely accessible to all readers. They include a short stand first (maximum of 300 characters, including space) as well as 2-5 one-sentences bullet points that summarizes the paper. Please write the bullet points to summarize the key NEW findings. They should be designed to be complementary to the abstract - i.e. not repeat the same text. We encourage inclusion of key acronyms and quantitative information (maximum of 30 words / bullet point). Please use the passive voice. Please attach these in a separate file or send them by email, we will incorporate them accordingly.

Please also suggest a visual abstract to illustrate your article as a PNG file 550 px wide x 300-600 px high. A cropped portion of this image will serve as thumbnail for the table of content on our webpage.

14) As part of the EMBO Publications transparent editorial process initiative (see our Editorial at <http://embomolmed.embopress.org/content/2/9/329>), EMBO Molecular Medicine will publish online a Review Process File (RPF) to accompany accepted manuscripts.

In the event of acceptance, this file will be published in conjunction with your paper and will include the anonymous referee reports, your point-by-point response and all pertinent correspondence relating to the manuscript. Let us know whether you agree with the publication of the RPF and as here, if you want to remove or not any figures from it prior to publication. Please note that the Authors checklist will be published at the end of the RPF.

I look forward to receiving your revised manuscript.

Yours sincerely,

Lise Roth

We thank the reviewers for their valuable comments on our EMBO Molecular Medicine Report manuscript. Two of the reviewers raised concerns regarding the lack of novelty in our findings vis-à-vis the work from the Long laboratory and the small number of patients profiled in our study.

The studies from the Long laboratory (Li et al, 2022; Xiao et al, 2024) profiled Lac-Phe in sera from normal subjects (following exercise) and diabetic non-oncologic patients (following metformin treatment). To the best of our knowledge at the time of manuscript submission, there are no published studies assessing Lac-Phe in any cancer setting (let alone in patient samples). This is an important consideration as dysregulation of metabolic pathways is a well-known hallmark of cancer. At present there is no evidence, a priori, that the Lac-Phe generating pathway and its stimulation by metformin would operate similarly in cancer as it does in diabetic patients and healthy humans. As Reviewer 1 correctly notes, our finding speaks to the generalizability of the metformin/Lac-Phe connection to cancer and, as such, is important to the medical field, given the commonly prescribed nature of metformin. Finally, we report absolute Lac-Phe concentrations in our study. These levels are consistent with those induced by and correlated to the metabolic benefits of exercise (Li et al. Nature, 2022), of which some cancer patients may not be capable. Thus, the potential for metformin recapitulating these benefits of exercise in the cancer context is an important and novel implication raised by our study. While our studies build on the earlier published work in the non-oncologic context, we do not believe our study is confirmatory as it adds key oncologic context in a disease (prostate cancer) where the standard-of-care (hormonal therapies) leads to significant metabolic dysfunction and weight gain, increasing the risk of adverse cardiovascular events. These points are now explicitly included in our manuscript in the abstract and the main text.

Our studies used all the available samples (12 out of 29 originally enrolled patients) from BIMET-1, a prospective prostate cancer metformin trial, enabling comparisons between controlled patient cohort arms. Reviewers 2 and 3 raised the issue of why we only profiled 12 samples – we have now clarified in revised text that these were all the samples available to us. For context, the 2024 Nature Metabolism paper that originally reported the metformin/Lac-Phe link profiled 21 retrospective Type 2 diabetic patient samples from the original cohort of 31 patients. This study also profiled retrospective samples from 1184 out of 6814 participants in the community-based MESA study, which was intended to assess the effects of metformin use on microvascular complications in diabetic patients, reporting a small increase in normalized relative Lac-Phe levels. While these studies have larger sample numbers, the subjects typically take variable metformin doses, and other medications are not controlled for in the study. The BIMET-1 patients we profiled were enrolled in one of the very few prospective metformin trials in prostate cancer and were thus on a controlled and monitored metformin dose.

To assess the robustness of the metformin/Lac-Phe link in prostate cancer and in response to the reviewer comments regarding small study numbers, we have now added profiling of sera from 25 additional prostate cancer patients recruited to an umbrella specimen collection protocol. Given how commonly metformin is prescribed, we reasoned some subset of the patients whose samples were available may be taking metformin. Moreover, unlike BIMET-1, these patients represented metastatic hormone-sensitive, biochemically recurrent and castration-resistant disease stages with several patients on hormone therapy. The profiling in this second cohort was conducted initially blinded to the patient's metformin status, and we found that the patients clustered into high and low Lac-Phe groups.

Following unblinding, remarkably, we found all patients on metformin (at doses of either 500 mg or 1000 mg) fell into the high Lac-Phe group (new Fig. 1F). Moreover, our results in the newly added patient cohort (Fig. 1F) suggest other metabolism-modifying interventions such as tirzepatide and semaglutide should also be evaluated for their ability to raise Lac-Phe levels. Notably, none of the patients in the low Lac-Phe group were on any metabolism-modifying treatment. Thus, this second cohort supports the robust and broadly generalizable link between metformin use and Lac-Phe induction, independent of prostate cancer stage, hormone therapy status, and other patient variables (see new Fig. 1F, Fig. EV1C). It further raises the possibility that Lac-Phe may be a common molecular metabolic nexus for other glucose control/anti-obesogenic agents.

Finally, we have now significantly rewritten the abstract and manuscript text to address these concerns and other reviewer critiques. We have also streamlined the data figures to more clearly and transparently present our findings. Detailed responses to reviewer comments are provided below.

Referee #1

This is a technically well done study. The novelty is considered medium because this is a replication and extension of the original Xiao et al Nature Metabolism paper, now in a new cohort of prostate cancer patients. The medical impact is high because it is important to understand the effect of metformin on these patients, and how general the metformin/Lac-Phe connection actually is. This is a well done study that shows metformin-inducible increases in both Lac-Phe and GDF15 in a small cohort of prostate cancer patients. The technical mass spec aspects of this study are well done. The patient cohort is well characterized. The authors also extend the analysis to GDF15, which is good and important because these two molecules seem to both be inducible by similar stimuli including metformin. In my opinion this is suitable for publication.

We thank this reviewer for their support of our publication and for their appreciation of a key point in our manuscript, namely the generalizability of the metformin/Lac-Phe connection to the cancer setting, which is often characterized by dysregulation of metabolic pathways.

Referee #2

Major points:

(1) The major issue of this study is the small sample size and large interpersonal variations. Among 12 patients, two serum samples are not available.

While we appreciate that the sample sizes in our study are small, we evaluated all the available samples from the BIMET-1 trial, one of few prospective trials of metformin in cancer and particularly in prostate cancer. Given that these are patient-derived specimens, we find it unrealistic to not expect interpersonal variation in profiling of metabolites. Indeed, this is also the case in the other studies profiling Lac-Phe in non-oncologic cohorts (Li et al, Nature 2022; Xiao et al, Nature Metabolism, 2024) and the inter-individual variations seen in our study are comparable or lesser than those seen in these other studies. More importantly, the metformin/Lac-Phe link is extremely consistent in all our profiled samples.

As discussed above, blinded profiling of Lac-Phe levels in the additional 25 prostate cancer patient samples across the disease spectrum clearly shows that metformin treatment significantly elevates Lac-Phe levels in prostate cancer patients independent of disease stage or their hormone therapy status. Thus, our collective data in Fig. 1 (now in a total of 37 patient samples) support the robust link between metformin use and Lac-Phe elevation in the cancer setting.

(2) while the body weight changes by metformin treatment is rather mild with large variations (only a small portion of patients shows a modest weight loss from Figure 2A), increases in Lac-Phe are detected in almost all the patients. In this connection, the data presented in panel A, C and D of figure 2 appear to be contradictory.

We have now emphasized in the manuscript text that the critical issue for cancer patients undergoing hormonal therapies is one of weight management i.e. lack of significant weight gain, and not weight loss per se, which can be undesirable in some cancer patients. We have simplified our data presentation in Fig. 2. As can be seen from the revised version of Fig. 2A, the Arm B (metformin) cohort shows overall better weight management compared to the Arm A patients following six months of anti-androgen (bicalutamide) therapy. It also shows that all Arm B patients lost weight on metformin monotherapy, congruent with all Arm B (but not Arm A) patients showing an increase in Lac-Phe levels.

In the Nature Metabolism paper (Xiao et al, 2024), the retrospective analysis of the MESA 1184 participant cohort required post-hoc bootstrapping mediation analysis to find significant correlations among Lac-Phe levels, metformin use, and weight loss. No significant correlation was noted for weight gain. However, even in this large group analysis, including Lac-Phe in the unmediated model weakened the correlation between metformin and weight loss, suggesting there may be additional lifestyle, dietary and/or molecular factors that influence the Lac-Phe/weight loss correlation. Despite the small numbers in our study, the trends we see between metformin

treatment, weight and Lac-Phe levels in the BIMET-1 patients are consistent with the conclusions of Xiao et al. study that metformin elevates Lac-Phe and that this elevation mediates metformin's anti-obesogenic effects.

(3) In Figure 3E, the lack of significant correlation between the changes in body weight and GDF15 might be due to the small sample size. The data cannot be used to exclude the involvement of GDF15 in metformin-mediated weight reduction

We have now placed the correlations between changes in weight vs. those in Lac-Phe or GDF-15 levels as side-by-side subfigures within the same figure (Figs. 2D, E). The text now makes it clear we are showing trends rather than significant differences.

Despite the significantly increased GDF-15 levels in Arm B patients on metformin monotherapy (Figs. 2B, C), there appears to be no trend of association whatsoever between weight loss and changes in GDF-15 levels (Fig. 2E; $p=0.7785$, Pearson coefficient = 0.1316). In comparison, the trend of association between weight loss and changes in Lac-Phe levels in the same metformin monotherapy patients is much stronger (Fig. 2D, Pearson coefficient = -0.6214). Therefore, while neither correlation is statistically significant, our overall data as well as prior findings support the likelihood that Lac-Phe, rather than GDF-15, mediates metformin-induced weight loss. We have altered our original text as follows: *'Thus, while not conclusive, our results support findings from other studies that report elevated Lac-Phe underlies metformin-induced weight loss and that, despite its increased levels, the GDF-15 axis is likely to be dispensable for metformin-induced weight loss (Klein et al, 2022; Xiao et al., 2024).'*

(4) Another limitation of this study is that only male patients are included. It remains unclear whether the observed effect of metformin on Lac-Phe also occurs in females.

Our study focused on prostate cancer as obesity and metabolic dysfunction contribute to treatment failure and disease progression and, moreover, raise the risk of hormone therapy-induced fatal cardiovascular events. Thus, weight management and metabolic status are significant determinants of disease mortality and treatment outcomes in prostate cancer. Moreover, we had access to samples from BIMET-1, a unique prospective metformin trial that enrolled only overweight or obese patients, making it an ideal cohort in which to evaluate effects of metformin on Lac-Phe and weight management. Understandably, in this setting, no female patients were available. We have now added this caveat to the final sentence of our manuscript: *'More broadly, with the natural caveat that our study does not include female cancer patients, our results generalize the metabolic link between metformin and Lac-Phe first observed in T2D patients and healthy individuals (Li et al., 2022; Xiao et al., 2024) to the oncologic context.'*

In the Paper Summary text, we also now include this caveat: *'It remains to be seen if this pathway also operates in other cancer patients, particularly female patients.'*

(5) As this study only shows the clinical association between changes in Lac-Phe and body weight, it does not support the conclusion that Lac-Phe protects against treatment-induced weight gain (last sentence of abstract).

We agree the last sentence of our original abstract was an overreach based on our data and we have now removed it. We have also overall softened this conclusion in the text of the manuscript. We thank the reviewer for pointing out this important shortcoming of our original manuscript.

Referee #3 (Remarks for Author):

The article by Bilusic et al. presents the results of an ancillary analysis of the BIMET-1 clinical trial, a randomized phase 2 study investigating the effects of combining metformin and bicalutamide on prostate cancer recurrence in obese patients. The focus of this ancillary study is N-lactoyl phenylalanine (Lac-Phe), a gut-derived metabolite previously reported to be induced by exercise. The authors seek to determine whether Lac-Phe levels are elevated in patients and whether it can mimic the effects of physical activity. The sole novel finding of this manuscript is the observed increase in serum Lac-Phe concentrations in patients treated with metformin. However, the effects of metformin on weight management have already been documented in the STAMPEDE trial, and there is no significant

correlation between weight loss and Lac-Phe concentration, suggesting that these parameters are not linked.

Lac-Phe is an exercise-induced anti-obesogenic metabolite. The main point of our study is the generalizability of the link between metformin use and Lac-Phe elevation in the cancer context, given the potential for its safe oncologic repurposing. The BIMET-1 trial, from which some of our samples derive, is one of few prospective metformin studies in prostate cancer and, unlike the larger trial STAMPEDE, it specifically enrolled only overweight or obese (but non-diabetic) patients. We analyzed the 12 samples available to us from this prospective metformin trial. Nevertheless, we found an extremely consistent effect of increased Lac-Phe with metformin treatment in this cohort of patients, even after they were placed on the anti-androgen, bicalutamide. These patients had significantly better weight management than the patients on bicalutamide alone (new Fig. 2A, Arm B vs. Arm A). The patients on bicalutamide alone did not show significantly increased Lac-Phe levels (Fig. 1C) and correspondingly greater net weight gain (Fig. 2A). This is an important finding as prostate cancer is a specific example of a malignancy where the standard of care, hormone blockade, can induce significant metabolic dysfunction and weight gain.

Major Points:

- **The authors should include a schematic overview of the clinical trial design.**

We thank the reviewer for this suggestion and have now included a schematic in Fig. 1A, showing the treatment arms and the number of patient samples available for profiling in this study.

- **While the original BIMET-1 trial enrolled 29 patients, this ancillary study includes only 12 patients. The authors need to address this discrepancy.**

As discussed above, we profiled all available samples from the BIMET-1 study – this has now been mentioned in the text. As with any prospective clinical study, the samples were limited as they have been shared with other stakeholders for separate correlative analyses.

- **In Figure 2D, the conclusion does not align with the data; the correlation between weight loss and Lac-Phe concentration is not statistically significant.**

We have now simplified and streamlined the new Fig. 2. We note all Arm B BIMET-1 patients showed an increase in Lac-Phe on the metformin monotherapy arm (Fig. 1D) and all lost weight compared to baseline (Fig. 2A, EV2A). Although the correlation between net weight loss and changes in Lac-Phe is not statistically significant ($p=0.1363$), it is quite strong (Pearson $r = -0.6214$). By comparison, despite the significantly increased GDF-15 levels in Arm B patients on metformin monotherapy, there appears to be no trend of association whatsoever between weight loss and changes in GDF-15 levels ($p=0.7785$, Pearson $r = 0.1316$). We have now softened our analysis in the manuscript text as discussed above in our response to Reviewer 2 (point 3).

- **Overall, the study's small sample size limits the validity of its conclusions, and the authors overinterpret the findings.**

While we appreciate that the sample sizes in our study are small, we evaluated all the available samples from the BIMET-1 trial, one of very few prospective trials of metformin in prostate cancer. As discussed above, since the original submission, we acquired and profiled another 25 prostate cancer patient samples across the disease spectrum for serum Lac-Phe concentrations. This profiling clearly shows that metformin treatment significantly elevates Lac-Phe levels in patients across prostate cancer disease stages, hormone therapy status and other patient variables (Fig. 1F, Fig EV1C), supporting the robust link between metformin use and Lac-Phe elevation in the cancer context. We have also significantly rewritten the text to better align our conclusions with our data.

Conclusion: This study is overly preliminary and does not provide substantial new evidence regarding the impact of metformin in patients with prostate cancer.

To the best of our knowledge, there are no published studies assessing Lac-Phe in any cancer setting (let alone in patient samples). This is an important consideration as metabolic dysregulation is a well-known hallmark of cancer. At present there is no evidence, a priori, that the Lac-Phe generating pathway and its stimulation by metformin would operate similarly in cancer as it does in diabetic patients and healthy humans. Thus, our findings establish, for the first time, that metformin-induced elevation of Lac-Phe occurs in prostate cancer patients, extending a previously described metabolic mechanism from non-oncologic populations into the cancer setting. Indeed, as Reviewer 1 correctly notes, our main finding speaks to the generalizability of the metformin/Lac-Phe connection to cancer and, as such, is important to the medical field, given the commonly prescribed nature of metformin and its ability to exert glucose and weight control without compromising oncologic care.

Our study raises some new possibilities. We report absolute Lac-Phe concentrations in our study, which is important for any clinical follow-up or further biomarker analysis. These levels are consistent with those induced by and correlated to the metabolic benefits of exercise (Li et al. *Nature*, 2022) of which some cancer patients may not be capable. Thus, the potential for metformin recapitulating these benefits of exercise in the cancer context is a new and important hypothesis raised by our study. Finally, data from our non-trial patient cohort (with the caveat of small numbers) raises the possibility that anti-obesogenic agents, semaglutide and tirzepatide, may also exert their benefits through the Lac-Phe axis. Thus, while our studies build on the earlier published work in the non-oncologic context, we do not think they are confirmatory as they add significant and novel context.

Finally, it is unclear why Reviewer 3 considers our study preliminary - we specifically opted for the Report format because our study focuses on a single consistent and novel finding, established through rigorous state-of-the-art analyses in prostate cancer patient samples from a small prospective trial and now in additional patients from a biomarker specimen collection protocol. To the best of our understanding, our study is consistent with the requirements of a Report format article as defined by the journal.

6th Feb 2026

Dear Dr. Rai, Dear Priya,

Thank you for submitting your revised study. Please accept my apologies for the delay in responding, which was due to reduced office activity over the holiday period and the high volume of new submissions received during that time.

We have now received the reports from the three initial referees. As you will see below, referees #1 and #3 are satisfied with the revisions and support publication. However, referee #2 remains critical of the sample sizes and of the associative nature of the study. Having discussed the manuscript and referee reports further within the team, we have agreed that, given the potential medical interest of the findings and the positive reports of referees #1 and #3, we would like to invite you to make the following minor revisions to the manuscript:

1/ Please address referee #2's concerns by clearly stating the limitations of the study.

2/ Manuscript text:

- Please indicate in track changes mode any new modification to the text.

- I would suggest including "Prostate cancer patients" in the title.

- "Material and Methods" should be renamed "Methods". The information provided should match the information provided in the author checklist:

o If study protocol has been pre-registered, provide DOI in the manuscript. For clinical trials, provide the trial registration number OR cite DOI.

o Report the clinical trial registration number (at ClinicalTrials.gov or equivalent), where applicable.

o If collected and within the bounds of privacy constraints report on age, sex and gender or ethnicity for all study participants.

o Please include a statement confirming that the experiments conformed to the principles set out in the WMA Declaration of Helsinki and the Department of Health and Human Services Belmont Report.

o Statistics: Please provide information on randomization.

- Data availability: please remove "The source data for the figures are provided as Supplementary materials."

- Acknowledgments: Please ensure that the funders list in our system is complete and accurate (it should match the information provided in the manuscript). The Women's Cancer Association (WCA) cancer research award (PR) and Sylvester Comprehensive Cancer Center research development funds, Florida Department of Health (24B12), Melanoma Research Alliance (1434401), and the Miami VA Healthcare System GRECC are missing in our system. Please ensure that all funders are listed, as this will be linked to PubMed upon publication. Please do not enter any information into the comments field, as this cannot be linked.

3/ Figures and Appendix:

- Appendix: Please add page numbers in the table of contents.

- Please correct the nomenclature of the expanded view figures to "Figure EV1" and "Figure EV2" and add the heading "Expanded View Figure Legends" to the manuscript text file.

- Please address the query from our data editors in the figure legends:

Please note that information related to n is missing in the legends of figures 1B, C; 2A, B.

4/ Thank you for providing Source Data. The files should be zipped to have one file per figure.

5/ The paper explained:

Please include it in the manuscript text file, and correct the heading and format to "The Paper Explained", followed by 3 subsections: "Medical Issue - Results - Clinical Impact".

Please refer to any of our published articles for an example.

6/ Synopsis:

I rephrased the stand first of your synopsis, please let me know if you agree or amend as you see fit. The subsequent bullet points should summarize your findings, please rephrase accordingly. Please refer to any of our published articles for an example.

Stand first:

"The anti-obesogenic metabolite Lac-Phe was identified as a potential mediator of the metabolic benefits of metformin use in cancer patients. This has implications for improving long-term outcomes in patients undergoing therapies that significantly increase cardiometabolic risk."

Thank you for providing a nice visual abstract. Please upload it as a png/tiff/jpeg file 550 px wide x 300-600 px high. If you used an image database for scientific iconography (e.g., BioRender), please let us know if you have a license that allows for publication in an academic journal.

7/ As part of the EMBO Publications transparent editorial process initiative (see our Editorial at <http://embomolmed.embopress.org/content/2/9/329>), EMBO Molecular Medicine will publish online a Review Process File (RPF) to accompany accepted manuscripts.

This file will be published in conjunction with your paper and will include the anonymous referee reports, your point-by-point response and all pertinent correspondence relating to the manuscript. Let us know whether you agree with the publication of the RPF.

I look forward to receiving your revised manuscript.

With kind regards,

Lise

Lise Roth, PhD

Senior Editor

EMBO Molecular Medicine

***** Reviewer's comments *****

Referee #1 (Remarks for Author):

This manuscript is appropriate for publication

Referee #2 (Remarks for Author):

The authors attempted to address my concerns by rewriting the abstract and manuscript text. However, the issues of small sample sizes and association nature of this study remain to be a major concern.

Referee #3 (Comments on Novelty/Model System for Author):

The manuscript has been improved, but the novelty remains moderate, though the study is still of significant interest.

Referee #3 (Remarks for Author):

The manuscript has been significantly improved since its first version, and the inclusion of additional patient data strengthens both the conclusions and the overall impact of the study. The authors have addressed my comments thoroughly, and I now consider the manuscript suitable for publication.

Referee #1

We thank this reviewer for their support of our manuscript and for their appreciation of a key point in our manuscript, namely the generalizability of the metformin/Lac-Phe connection to the cancer setting, which is often characterized by dysregulation of metabolic pathways.

Referee #2

The authors attempted to address my concerns by rewriting the abstract and manuscript text. However, the issues of small sample sizes and association nature of this study remain to be a major concern.

We appreciate the feedback from this reviewer in the prior revision cycle, which has helped us improve our manuscript. We increased our sample size by including Lac-Phe profiling from 25 additional PC patients. Based on this reviewer's feedback, we have also modified our language throughout the abstract and manuscript to be clear about the small numbers in our study (for example, with regards to data in Fig. 1) and about any results that lack significance but rather suggest a trend of association (for example, with regards to data in Fig. 2). For transparency regarding the sample size we evaluated, we have included the specimen numbers from the BIMET-1 trial in all figure legends. All related graphs are presented with individual patient values plotted, rather than box-and-whisker plots. We have also explicitly included the caveat raised by this reviewer that by focusing on prostate cancer, a disease where standard-of-care has significant cardiometabolic risks, we have not captured whether metformin also raises Lac-Phe to similar levels in female cancer patients. Our title now includes the word 'prostate cancer' to emphasize the specificity of our disease model and its limitations. Nevertheless, our results in the revised manuscript support a robust link between metformin use and elevated Lac-Phe across the spectrum of high-risk PC patients, including those with castration-resistant disease, even when we cannot control for specific metformin dose, co-existing metabolic dysfunction or hormone therapy status. Thus, we believe our study has important clinical and translational implications for metformin use in oncology, beyond its anti-cancer efficacy. Given the relative paucity of studies evaluating the metformin/Lac-Phe link in any cancer setting, let alone in patient samples, we believe that our results will be of interest to basic and clinical scientists working on cancer metabolism and on lifestyle interventions to improve cancer outcomes.

Referee #3 (Remarks for Author):

We thank this reviewer for their support of our revised manuscript and for their helpful comments in the prior revision cycle, which have strengthened and clarified the premise of our study.

3rd Mar 2026

Dear Dr. Rai, Dear Priya,

Thank you for submitting your revised files. I am pleased to inform you that your manuscript is accepted for publication and is now being sent to our publisher to be included in the next available issue of EMBO Molecular Medicine.

You may qualify for financial assistance for your publication charges - either via a Springer Nature fully open access agreement or an EMBO initiative. Check your eligibility: <https://link.springer.com/journal/44321/how-to-publish-with-us>

With kind regards,

Lise

>>> Please note that it is EMBO Molecular Medicine policy for the transcript of the editorial process (containing referee reports and your response letter) to be published as an online supplement to each paper. If you do NOT want this, you will need to inform the Editorial Office via email immediately. More information is available here: <https://link.springer.com/partners/embo-press/editorial-policies#Peer%20review>